# The Effects of Non-Pharmacological Interventions in Fibromyalgia: A Systematic Review and Metanalysis of Predominants Outcomes

**DOI:** 10.3390/biomedicines11092367

**Published:** 2023-08-24

**Authors:** Isabel Hong-Baik, Edurne Úbeda-D’Ocasar, Eduardo Cimadevilla-Fernández-Pola, Victor Jiménez-Díaz-Benito, Juan Pablo Hervás-Pérez

**Affiliations:** 1Department of Physiotherapy, Faculty of Health, Camilo José Cela University, 28692 Villanueva de la Cañada, Madrid, Spain; isabel.hong@alumno.ucjc.edu (I.H.-B.); eubeda@ucjc.edu (E.Ú.-D.); ecimadevilla@ucjc.edu (E.C.-F.-P.); 2Department of Sport Sciences, Faculty of Physical Activity and Sport Sciences, Universidad Europea de Madrid, 28670 Villaviciosa de Odón, Madrid, Spain; victorjimenezdb@gmail.com

**Keywords:** cytokines, fibromyalgia, physiotherapy, metanalysis

## Abstract

(1) Fibromyalgia (FM) is a chronic musculoskeletal condition with multiple symptoms primarily affecting women. An imbalance in cytokine levels has been observed, suggesting a chronic low-grade inflammation. The main aim of the meta-analysis was to examine the effect of multimodal rehabilitation on cytokine levels and other predominant variables in patients with FM. Furthermore, to examine which non-pharmacological tools have been used to investigate the effects that these can have on cytokines in FM patients. (2) Methods: Searches were conducted in PubMed, Scopus, Web of Science, Cochrane, and ScienceDirect databases. This systematic review and metanalysis followed the PRISMA statement protocol. The methodological quality of the studies was assessed using the PEDro scale, the risk of bias followed the Cochrane Manual 5.0.1, and the GRADE system was used for rating the certainty of evidence. (3) Results: Of 318 studies found, eight were finally selected, with a sample size of 320 women with a mean age of 57 ± 20. The proinflammatory cytokines IL-1β, IL-6, IL-8 and TNF-α were the most studied. Resistance exercise, aquatic exercise, dynamic contractions, cycling, treadmill, and infrared therapy were the main non-pharmacological tools used. (4) Conclusions: The systematic review with meta-analysis found evidence of elevated cytokine levels in patients with FM, suggesting low chronic inflammation and a possible contribution to central sensitization and chronic pain. However, the effects of physiotherapeutic interventions on cytokine levels are variable, highlighting the importance of considering different factors and the need for further research.

## 1. Introduction

Fibromyalgia (FM) is characterized by chronic, widespread musculoskeletal pain with multiple tender points and generalized tenderness with muscle stiffness, joint stiffness, sleep disturbances, fatigue, mood, cognitive dysfunction, anxiety, depression, general tenderness, and inability to perform daily life activities. Regarding prevalence, it is estimated that 4% of the world’s population is affected by FM, mainly in women aged 20–55 years [1].

The diagnosis is only clinical and consists of a complete assessment based on the 1990 American College of Rheumatology (ACR) criteria of three consecutive months of widespread pain and “tender points” of pain on palpation. In 2010, the ACR updated the criteria with two new parameters, and in 2016, the criteria were further revised to decrease the likelihood of misdiagnosis [2].

FM is a syndrome with a multifactorial etiology that develops depending on genetic predisposition, personal experiences, emotional and cognitive factors, and the individual’s ability to cope with stress [3].

Although FM is traditionally a non-inflammatory condition, current evidence suggests that other factors contribute to its pathogenesis, such as inflammatory, immunological, and endocrine factors [4]. In FM, there is increasing evidence of inflammatory mechanisms of neurogenic origin in peripheral tissues, the spinal cord, and the brain. Cytokines/chemokines, lipid mediators, oxidative stress and various plasma-derived factors underlie the inflammatory state in fibromyalgia [1].

This inflammation is closely related to the activation of both the innate and adaptive immune systems that produce an inflammatory cascade of neuropeptides, cytokines, and chemokines, which may play an essential role in the pathophysiology of FM [5]. Cytokines function as messengers of the immune system and have a regulatory role in inflammation and are essential in a brief form; the problem comes when there is prolonged exposure to them, leading to chronic low-grade inflammation. Cytokines are classified into proinflammatory and anti-inflammatory cytokines. The central proinflammatory cytokines are IL-1, IL-2, IL-6, IL-8, IL-12, TNF-α, and IFN, while the central anti-inflammatory cytokines are IL-4, IL-10, IL-13, and TGF. The proinflammatory to anti-inflammatory cytokines ratio is vital in determining disease outcomes [4]. Current research has shown that cytokine levels are imbalanced in the human body and that the proinflammatory to anti-inflammatory cytokines ratio is critical in determining disease outcomes [4].

Current research has shown that cytokine levels are imbalanced in FM patients. There is an imbalance between proinflammatory and anti-inflammatory cytokines, with more proinflammatory cytokines such as TNF-α, IL-1RA, IL-6, and IL-8 [6].

Various non-pharmacology tools and their efficacy in treating patients with FM have been examined in the scientific literature, including strength exercise, aerobic exercise, aquatic exercise, and flexibility exercise. However, the latter is used as part of the warm-up. Therapeutic massage has also been studied, and techniques such as myofascial release, connective tissue massage, lymphatic drainage, and Shiatsu have been explored. In addition, electrotherapy has been incorporated into treatment protocols due to its efficacy and safety profile, including techniques such as Transcutaneous electrical nerve stimulation, transcranial magnetic stimulation, transcranial direct current, and laser [7]. Other tools that have been evaluated include balneotherapy, dry needling, acupuncture, and patient education. Current recommendations for the treatment of FM stress the importance of including patient education and the use of physiotherapy tools as part of the primary treatment [8]. Research on physiotherapy tools is aimed at analyzing whether they can generate effects on proinflammatory and anti-inflammatory cytokine levels in patients with FM [9,10].

This set of specific inflammatory and biochemical markers could help diagnose FM, so research is directed toward studying inflammatory markers to achieve a more objective diagnosis of FM [2]. Recent reviews with metanalysis [11] revealed that individual biomarkers may play a relevant role in identifying pathologies coexisting with FM. However, new research could make it possible to offer the predominant instruments and parameters depending on the type of intervention carried out. Once these parameters were classified, a grouping of predominant outcomes was carried out from the original data of the articles whenever these data were offered. The main aim of the meta-analysis was to examine the effect of multimodal rehabilitation on cytokine levels and other predominant variables in patients with FM. Additionally, the principal purpose of the present study was to analyze women with FM since, at present, the rate of diagnosis is still higher in women than in men, which undoubtedly represents a limitation to being able to draw conclusions about this disease in the male gender.

## 2. Materials and Methods

### 2.1. Acquisition of Evidence

For this systematic review, we followed the protocol according to the standards and guidelines of the PRISMA statement [12] for systematic reviews and meta-analyses, which aims to improve the reporting of future systematic reviews. The methodological protocol was registered after the present work (protocol number: INPLASY202370033).

### 2.2. Eligibility Criteria

The components described in the PICO framework were applied to achieve appropriate results (P—patients with FM according to ACR 1990; I—physiotherapy intervention tool; C—comparison of healthy group vs. FM group (same intervention). FM group vs. FM group (physiotherapy and relaxation intervention); O—results … levels of proinflammatory and anti-inflammatory cytokines. In the control group and experimental group.

Articles that met the following inclusion criteria were included: (a) publications in the last ten years (from 2013 to 2023); (b) written in English or Spanish; (c) clinical trials and randomized controlled clinical trials using a placebo or control group; (d) subjects with a diagnosis of FM according to the ACR 1990; (e) women of working age (an intervention group performing a physiotherapeutic intervention; (f) cytokine analysis. Exclusion criteria were: (i) subjects with pathologies other than FM; (ii) clinical trials with no results or not completed.

### 2.3. Sources of Information

The databases Pubmed, Scopus, Web of Science, Cochrane Register, and ScienceDirect were searched. A search was also made in second-line sources, doctoral theses, journal articles, etc., in Dialnet; and Teseo, but there were no results.

These searches were carried out from December 2022 to March 2023.

### 2.4. Search Strategies

Before starting the present systematic review, a search of different databases was carried out to verify the existence of recent reviews on the topic in question. Subsequently, several searches were performed using different combinations of the key terms, including “cytokines AND fibromyalgia” and “cytokines AND fibromyalgia AND physiotherapy”. It used the search equation (“cytokines and Fibromyalgia and Physiotherapy”) to focus the search on studies that used physiotherapy tools as an intervention and analyzed their possible effects on cytokines in FM patients. The same Medical Subject Headings (MeSH) terms were used to improve the specificity of the search.

A series of filters were established and included in each database to perform the current searches, which are detailed in Section 2.2 eligibility criteria.

The flow diagram below graphically shows the selection of the articles included in this systematic review and metanalysis.

### 2.5. Study Selection Process

In the present systematic review and metanalysis, the assessment of potentially relevant studies was performed based on their title and abstracts. The independent variables included in each of the selected studies were the number of patients, the number of withdrawals in each group, the clinical variables of the participants, the measurement tools evaluated, and the characteristics of the intervention applied. The primary dependent variable analyzed was the presence of proinflammatory and anti-inflammatory cytokines in plasma and muscle, regardless of the tool used to measure them. The secondary variables analyzed were the non-pharmacological techniques used. In total, eight studies were included in the systematic review and metanalysis.

### 2.6. Data Extraction Process

An exhaustive reading and evaluation of the eight studies finally selected were carried out, to which the PEDro scale was applied to assess their methodological quality, evaluating the design of the study, the source of obtaining the subjects, whether the study was randomized, whether there was concealment, whether there was blinding and what the outcome of the study was like. The PEDro scale of the synthesis results can be found in more detail in (Figure 1).

Also, the PRISMA 2020 checklist [12] was used to collect the most relevant data from each of the studies, author and year, type of study, sample characteristics, objectives, type of intervention, intervention time, number of sessions, frequency of sessions, session time, measures assessing the impact of FM, pain, fatigue, quality of life, depression, anxiety, physical capacity, blood tests to measure the different cytokines, results, conclusions, limitations of the study and the follow-up. The results of the data extraction will be presented in (Appendix A).

A reviewer working independently carried out both the study selection process and the data extraction process for each of the final articles. Subsequently, the results obtained were analyzed by two independent reviewers (IHB and EUD). In case of doubt or disagreement between the reviewers, they were jointly assessed until a consensus was reached.

### 2.7. List of Data

The (Appendix A) presents the table of the most studied cytokines, clearly identifying the most investigated and analyzed cytokines in the studies included in this review. On the other hand, (Appendix A) contains a comparative table of cytokine levels between the control group of healthy subjects and the experimental group of subjects with FM, which will provide a more accurate and understandable picture of the differences between the two groups.

### 2.8. Risk of Bias Assessment of Individual Studies

Risk of bias is a tool developed by the Cochrane Collaboration to assess the methodology of scientific evidence. It is useful in systematic reviews for the individual analysis of included CTs and RCTs. In this sense, the present systematic review has followed the Cochrane Handbook 5.1.0 [13] to assess the risk of bias.

The Cochrane Handbook 5.1.0 presents six levels of bias: selection bias, conduct bias, detection bias, attrition bias, reporting bias, and other bias. Each level has one or more specific items in a Risk of Bias table, and each item includes a description of what happened in the study and an assessment where the assignment of “low risk”, “high risk”, or “unclear risk” of bias is included [13]. The risk of bias assessment for each included study can be found in (Appendix A) of this systematic review.

### 2.9. Synthesis Methods

The synthesis methods used in the present review are the eligibility criteria that were determined in Section 2.2 of material and methods and the analysis of methodological quality using the PEDro scale, which is based on the Delphi checklist developed by Verhagen [14]. The checklist has a total of 11 items. The first item refers to external validity and is not considered for the final score; items 2–9 refer to internal validity, and items 10 and 11 indicate whether the statistical information provided by the authors allows for an adequate interpretation of the results.

Therefore, the maximum score is 10 points, and the minimum is 0. Only items that are answered affirmatively are scored. Studies with a score of 9–10 were of excellent methodological quality, 6–8 were of good quality, and 5 were of fair or acceptable quality. The PEDro scale can be found in more detail in Section 3.5 Results of the Synthesis.

Further to the synthesis measures, we assessed whether the studies included in the analysis met their objectives set at the start of the study. Of the eight studies included in this review, all of them met the objectives proposed at the outset. Regarding the homogeneity of the experimental and control groups of the studies, it was observed that in six of the eight articles [15,16,17,18,19,20], the groups were homogeneous, and the subjects were matched. However, in one of the studies [21], participants were not matched according to BMI, and the experimental group had higher BMI and blood pressure than the control group, and some participants had concomitant metabolic syndrome. In another study [22], the plasma analysis performed on the participants was not homogeneous, as the experimental group consisted of 75 subjects, while the control group had only 25 subjects. This information can be found in the descriptive tables in (Appendix A).

### 2.10. Assessment of the Certainty of the Evidence

The GRADE system [23] was followed to measure the assessment of the certainty of the evidence, which defines the quality of evidence as the degree of confidence we have that the estimate of an effect is adequate to make a recommendation. In classifying the quality of evidence, the GRADE system establishes four categories: high, moderate, low, and very low. From the present systematic review, five of the eight studies have a high quality of evidence [17,18,19,21,22], and three studies have a moderate quality [15,16,20]. These results can be seen in more detail in Section 3.6 Certainty of evidence.

### 2.11. Data Synthesis and Statistical Analysis

Statistical analysis of the meta-analysis was performed using the Review Manager software (RevMan 5.4; Cochrane Collaboration, Oxford, UK). A meta-analysis of the predominant variables using the same parameter and measurement scale was carried out using the random effects model, in which it was assumed that the effect of the treatments was not the same in all the studies included in the model. For this, the original values of each study (Mean and Standard Deviation or Median and interquartile range) were taken as reference. The effects of the experimental interventions against the comparison groups (controls, placebo, relaxation therapy, and healthy women without fibromyalgia) were presented as mean differences and their confidence intervals, taking a 95% CI as reference. The heterogeneity of the studies was evaluated using the I^2^ statistic, where values greater than 35% were heterogeneous. The variance between studies was calculated using Tau-square (Tau2) [24]. The significance level was set at 0.05 for statistically significant effects.

For the calculation of the effect size in cytokines, the predominant cytokine parameters at plasma levels (pg/mL) were selected and pooled from the original data of the different trials when a minimum N was obtained in two studies. Several trials reported median and interquartile ranges in their study. To combine the results of these studies, the mean and standard deviation of the sample were estimated using the method of Wan et al. [25]. This method, less effective than the method proposed by Hozo et al. [26], was more suitable for such an estimate. In cases where studies reported only *p* value greater than 0.05 or mean difference (Δ) in inter- and intragroup pre- and post-test value change, estimated values were taken from the original study data. Once all the necessary scores were obtained, the effect size of each parameter was calculated according to the method proposed by Lipsey and Wilson [27]. When the hypothesis testing of the original trial employed nonparametric techniques or more than two means were compared, Eta squared, or Eta squared partial (η^2^) was calculated directly or from Cohen’s d using the method of Lenhard et al. [28]. In some cases, the z-contrast statistic was calculated from the *p*-value offered in the trial. The cytokine meta-analysis model was random-effects based on the standardized mean difference and standard error of each study, also previously calculated [27]. The size of the effect on cytokines was considered small around 0.01, about 0.06 was considered a medium effect, and greater than 0.14 a large effect, this being negligible when 0 was found in the CI [29].

## 3. Results

### 3.1. Selection of Studies

During the initial stage of the search, 318 studies were identified from different databases. After removing duplicates, 302 studies remained.

To refine the selection, we applied date filters (2013–2023) and chose to select only clinical trials and randomized clinical trials, which were available in English or Spanish. After reviewing the titles and abstracts, 49 studies that did not fit the study topic were discarded, leaving ten studies for analysis. Of these, after a detailed reading, two studies were eliminated for inconclusive results, leaving a total of eight studies that met the inclusion criteria and were subjected to a qualitative analysis.

To carry out date and study type filters, we used database filters. To search for duplicates and perform the inclusion or exclusion of studies, we used an intelligent research collaboration platform called Rayyan, which optimizes efficiency in elaborating systematic reviews by facilitating the organization and classification of relevant studies to be considered.

### 3.2. Characteristics of the Studies

Of the total articles included in this review, 12.5% were published in 2013 [16], 25% in 2014 [15,20], 12.5% in 2015 [17], 12.5% in 2016 [18], 12.5% in 2018 [21] and 25% in 2019 [19,22]. A total of 575 individuals participated, all of whom were women with age ranges between 35 and 66 years; the median age was around 50. All FM patients were diagnosed by ACR 1990 criteria.

All articles were clinical trials or randomized clinical trials, presenting baseline measures, and comparing them with post-intervention. Of the eight studies included in this systematic review, seven present an experimental group including subjects diagnosed with FM and a control group composed of healthy subjects [15,16,17,18,20,21,22]; within these seven studies, there are two that also compare two types of interventions [exercise and relaxation] in subjects with FM [21,22]. A single study focuses on comparing the experimental group and a placebo group, while two studies compare two types of interventions (exercise and relaxation) in subjects with FM [19]. More detailed information can be found in the descriptive table for each of the studies in (Appendix A).

### 3.3. Risk of Bias in Individual Studies

A risk of bias assessment of the individual studies was performed, allowing a more accurate picture of the quality of the available evidence and the reliability of the results obtained. More detailed information on the risk of bias assessment of each of the studies included in this systematic review can be found in the tables in (Appendix A).

The risk of bias assessment figures for each study included in this systematic review are shown below. Each figure will show the result of the risk of bias assessment for each domain assessed, allowing us to identify the strengths and weaknesses of each study. In this way, we will gain a more detailed understanding of the quality of the included studies and their impact on the overall results of the systematic review.

In the risk of bias graph (Figure 2), it can be seen that in the blinding of participants and staff, blinding of assessors, incomplete short- and long-term outcome data, and selective reporting is 100% low risk whereas, in the generation of the randomized sequence, the risk of bias is 100% low, while randomized sequence generation, allocation concealment and other sources of bias are 70% low risk, 62.5% low risk and 37.5% high risk.

Furthermore, in the risk of bias summary (Figure 3), it can be seen which author and item has a low risk, unclear risk, and high risk. In the present review, clearly, those with the highest risk of bias are the non-randomized clinical trial studies [15,16,20], for the items of random sequence generation and allocation concealment, compared to the randomized clinical trial studies [17,18,19,21,22], which have in this case a low risk. Another item where a high risk of bias was found in three of the articles was in other sources of bias. In two of the studies [18,20], it was by allowing participants to continue taking the medication they were taking and controls to take medication during the intervention which could condition the results; in another study [21] it was by not matching subjects in the experimental group with subjects in the control group.

### 3.4. Results of Individual Studies

The results were obtained from the types of intervention; it was observed that different physiotherapy tools were used. Cycling was used in 12.5% [16]; aquatic exercise in 25% [15,19]; dynamic contractions in 25% [17,18,21]; strength exercise in 37.5% [18,21,22], a treadmill in 12.5% [20] and infrared in 12.5% [19].

In addition, each of the interventions used different durations and frequencies. They ranged from one-time sessions lasting 45 min, 20 min, or until exhaustion [16,17,20] to 6-week interventions, with three sessions per week, lasting 18–50 min per session [19], 15 weeks, with two sessions per week of 60 min [18,21,22] or eight months, with two sessions per week of 60 min per session [15]. Next, the most assessed variables were cytokine analysis, which was performed in all included studies [100%], followed by pain in 87.5% of the studies, fatigue, depression, physical capacity, and quality of life studied in 62.5% of the studies and anxiety and other variables in 50%. The tools used to measure variables such as FM were: FIQ [15,16,19,20,21,22] in 62.5%, FIQR [20] in 12.5%, SIQR [20] in 12.5%; the algometer (PPT) [18,20,21,22] was used to measure pain in 50% of the studies, VAS [17,18,19,22] in 50%, VAS [17,19,21] in 37.5%, PCS [21,22] in 25%, while SF-MPQ [19], MTPS [20], FTPS [20] and PDI in 12.5% of the studies; for fatigue, the Borg Scale [17,18] was used in 37.5% and the MFI [21,22] in 25%, for measuring the quality of life the SF-36 [16,17,18,21,22] in 62.5% and the SF-36-PSC [17,18,21,22], SF-36-MSC [17,18,21,22] in 50% of the studies. The HADS test [17,18,21,22] was used in 50% and the BDI-R [20] in 12.5% to measure anxiety and depression; for the measurement of physical capacity, the dynamometer [18,21] was used in 37.5%, the Rpar-Q [15,16] in 25%, the manual pressure force [21], the VO2 test [20], pulse rate [20] and blood pressure [20] in 12.5%; and the 6MWT [15,18,21] in 37.5% of the studies; for blood analysis muscle micro-dialysis [17,18] was used in 25% of the studies and venipuncture blood analysis in 100% [15,16,17,18,19,20,21,22]; other variables such as lifestyle, medication, and other diseases [15,16,20] were studied in 37.5% and finally, infrared thermography [19] in 12.5% of the studies.

Notably, the most studied variable was the blood test and cytokine expression before and after the intervention (in 100% of the studies). For this reason, a table was made to analyze the most studied cytokines in the studies we are concerned with here. In the evaluation of cytokines in blood obtained by venous puncture, it was observed that, of the 10 cytokines studied, seven were proinflammatory cytokines: IL-1β, IL-2, IL-6, IL-8, IL-17A, IL-18, and TNF-α, among these, IL-6 [16,17,18,19,20,21,22], IL-8 [15,16,17,18,20,21,22] and TNF-α [16,17,18,19,20,21,22] were studied in 87.5% of the studies, while IL-1β [16,17,18,20,21,22] was examined in 75% of them (Appendix A). IL-2 and IL-17A in 25% [21,22], and the least studied proinflammatory cytokine was IL-18 in 12.5% [16] of the studies. In contrast, only three anti-inflammatory cytokines were studied: IL-1ra, IL-4, and IL-10, with IL-10 being the most studied in 62.5% of studies [16,19,20,21,22], followed by IL-1ra in 37.5% [20,21,22] and IL-4 in only 25% of trials [21,22]. For muscle micro-dialysis, only the proinflammatory cytokine responses of IL-1β, IL-6, IL-8, and TNF-α were examined, and this measurement was performed in only 25% of studies [17,18] Appendix A. An additional table has been developed to compare cytokine levels between the control group of healthy subjects and the experimental group of subjects with fibromyalgia. In two studies [15,19], no data were obtained because the intervention was between groups of subjects with fibromyalgia. In two other studies [21,22], only a comparison of circulating cytokine levels between the control group and experimental groups was performed without applying any intervention, so we only have the baseline data. The most relevant finding in this table is that the cytokine levels in the control group of healthy subjects are lower than in the experimental group of fibromyalgia patients.

### 3.5. Results of the Synthesis

The articles included in the review were assessed using the PEDro methodological quality scale, shown below in Table 1. The final score obtained ranged from 5 to 10. Two studies were classified as being of excellent quality, three of good quality, and three of fair or fair quality. The studies achieved a mean value of 7.12 ± 2.88.

### 3.6. Certainty of Evidence

The GRADE system [23] has been used to classify the studies included in this review to determine the certainty of the evidence, five of the eight studies have a high quality of evidence [17,18,19,21,22], and three studies a moderate quality [15,16,20], and can be visualized more clearly in the following Table 2.

### 3.7. Metanalysis Results

Figure 4 shows the effects of the treatments versus the comparison group on the outcome measures. Ernberg et al. [20] showed a reduction of 7.1 points in the FIQ instrument. The treatments were effective in combination with the study by Slam et al. [19] [IV = −7.84 (−13.09 to −2.60); z = 2.93; *p* = 0.003]. Figure 5 shows the effects of the treatment compared to the comparison group on the algometry (PPT) in the left and right limbs. The study by Jablochkova et al. [21] found no statistically significant differences in the Algometer measurement in favor of the resistance program versus relaxation therapy (z = 0.19; *p* = 0.85), not observing intragroup effects on this variable in the original study. No heterogeneity was observed in either of the two analyses (χ^2^(1) = 0.89; I^2^ = 0% and (χ^2^(1) = 0.86; I^2^ = 0% respectively).

Likewise, a change of 2.5 points was found in the mental component of the SF-36 scale in favor of the group that underwent 15 weeks of progressive resistance exercise and in the overall combination in favor of treatment in the SF-36, the pain, and the variable Static strength knee extension force (Figure 6). No heterogeneity was observed in either of the FIQ, PPT, SF-36, and HADS analyses (I^2^ = 0%), however, heterogeneity was observed from χ^2^(2) = 196.33; *p* < 0.01; I^2^ = 99% for the pain variable; χ^2^(2) = 196.33; *p* < 0.01; I^2^ = 99%; for elbow flexion force and χ^2^(2) = 4.88; *p* = 0.09; I^2^ = 59% for the outcome knee extension force. Figure 6 shows the effects of treatment versus comparison group on the QoL (SF-36-PSC, SF-36-MSC and combined effects). Ernberg et al. [20] found a significant increase in the mental component of the quality of life of these patients after treatment (IV = −6.90 (−11.52 to −2.28)). The treatments were effective in combination with the study by Jablochkova et al. [21] (IV = −6.67 (−10.65 to −2.68); z = 3.28; *p* = 0.001). No statistically significant changes were found in the combination of values from the pre to post-test in quality of life in its physical dimension in any of the studies examined (*p* > 0.05) (Figure 7).

The studies by Ernberg et al. [17] and Jablochkova et al. [21] found a significant reduction in pain in favor of the experimental group IV = −20.00 (−22.64 to −17.36) and IV = 16.80 (−26.88 to −6.72); *p* < 0.05, respectively. However, no statistically significant differences were found in the overall analysis of both studies (Figure 8).

Regarding the effects of treatment versus comparison group on Static strength elbow flexion force, no statistically significant changes were found in the combination of values from pre to post-test in the results presented by right and left in any of the studies examined (*p* > 0.05) (Figure 9 and Figure 10).

Analysis of treatment effects was not significant in our IL-8 analyses. In the second study by Ernberg et al. [20], no intergroup effects were observed (Exercise treatment vs. Relaxaxion group) because 0 was found in IQ (*p* > 0.05), although there was a trend in favor of treatment in the original study (Figure 11).

No statistically significant differences were observed in the analysis of the effect of the studies on cortisol. Bote’s trial [16] found a difference of 0.38, but it was not significant (*p* > 0.05), as was the TNF variable (Figure 12 and Figure 13). The studies that examined the IL-1β variable found negligible effects in the trials included in the overall analysis conducted (*p* > 0.05) (Figure 14).

However, a statistically significant effect size was found in the study by Ernberg et al. [20] in favor of relaxation treatment on IL-6 (SMD = 0.50 (0.08 to 0.92; *p* < 0.05) (Figure 15).

## 4. Discussion

The main aim of the meta-analysis was to examine the effect of multimodal rehabilitation on cytokine levels and other predominant variables in patients with FM. The intervention by Ernberg et al. [20] significantly decreased the scores of the FIQ in favor of the relaxation group, but the two fibromyalgia groups examined did not find functional and clinical differences in the change in values from pre to post-test. Other studies showed differences in variables in favor of the control group in pain ratios applying programs of 15 weeks of strength exercise 2 sessions/week 1 session/60 min 10 min warm-up 50 min and lower limb strength, however, this intervention did not normalize a chronic inflammation profile nor did it have any effect on the anti-inflammatory effect in patients with FM symptoms on the clinical and functional variables examined [17,20,21].

Furthermore, the response of cytokine levels to intervention was found to differ between the groups. While in the healthy group, cytokine levels are low and increase during the intervention, in the experimental group, cytokine levels are elevated and decrease during the intervention. This observation suggests the presence of chronic inflammation in FM patients, which supports the hypothesis that inflammation and altered immune response may play a role in the development and maintenance of FM symptoms [4]. In addition, it also provides a possible explanation for many of the symptoms experienced by FM patients. For example, elevated concentrations of IL-6 and IL-8 have been shown to have additive or synergistic effects on the perpetuation of chronic pain in these patients [30]. IL-6 induces pain, fatigue, and psychiatric disorders such as depression and stress [4], while IL-8 is associated with pain and sleep disorders [16]. Consequently, an imbalance between proinflammatory and anti-inflammatory cytokines contributes to chronic peripheral sensitization of the nervous system and pain processing [2,4]. Another study also points to an imbalance between proinflammatory and anti-inflammatory cytokines, with higher levels of proinflammatory cytokines such as TNF-α, IL-1ra, IL-6, and IL-8 [6]. These results support the hypothesis that inflammation and immune responses are altered in the FM group.

Although the heterogeneity associated with the processes, techniques, and instruments for the detection of cytokines in different body fluids represents an underlying factor in the literature, we highlight the importance of using the mean difference technique. standardized in the meta-analysis. The study had difficulties in making Forrest plots regarding cytokines and the correlation with exercise in the study, as the authors did not show sufficient data to calculate the effect size from the original individual trial data. Our work showed a high methodological and statistical heterogeneity of the studies. The lack of data made it impossible to analyze all the variables, which is a limitation when it comes to quantifying the effects. This could affect the results of the meta-analysis, as some effects found were negligible in the meta-analytic model. However, a qualitative description of the effects of these variables has been made.

Proinflammatory cytokines are products of lymphocytes that are activated not only by injury but also by glial and neuronal cells. The main proinflammatory cytokines include IL-1, IL-2, IL-6, IL-8, IL-12, TNF-α, INF-α and IFN-α, while the main anti-inflammatory cytokines are IL-4, IL-10, IL-13, and TGF-α [4]. In the studies reviewed, mainly 10 cytokines were analyzed, with IL-1β, IL-6, IL-8, and TNF-α standing out as proinflammatory cytokines. However, there was a lack of investigation and analysis of anti-inflammatory cytokines, both at the muscle level [17,18], which only investigated the levels of proinflammatory cytokines, and at the plasma level where, for example, IL-13 was not analyzed in any of the studies.

According to investigate the cytokines present in the muscle tissue and plasma of patients with FM. In the present systematic review, only 25% of the studies [17,18] employed the micro-dialysis technique in muscle tissue and revealed no significant difference in the release of inflammatory cytokines between the groups in terms of IL-1β, IL-6, IL-8. However, an increase in the cytokine TNF-α was observed only in the control group. These findings align with the conclusions of other studies suggesting that physical exercise affects the metabolic profile of muscle but does not impact the plasma immune profile of FM patients [17,22].

The final specific objective was to examine the physical therapy tools used to investigate the effects of such interventions on cytokine levels in FM patients. In the studies that evaluated the effects of exercise on cytokines, variable results were found. Regarding the study of anti-inflammatory cytokines, we identified that in three studies, no analysis of these cytokines was performed [15,17,18]. In the other trials where they were analyzed, three found no evidence of an anti-inflammatory effect after the intervention, with no significant changes observed [20,21,22]. However, in three other studies, there was an anti-inflammatory effect [15,16,19]. Significantly, this effect was not directly attributed to the action of anti-inflammatory cytokines but to the reduction of pro-inflammatory cytokines, suggesting a possible inhibition of some anti-inflammatory cytokines, such as IL-10 [20]. For example, in one of the studies, after a moderate cycling session, an anti-inflammatory effect was found due to a slight decrease in IL-10 in the experimental group, but what made the difference was a significant reduction in IL-8 levels, attributed to a lower release of pro-inflammatory cytokines by monocytes and lower activation of neutrophils [16]. In another study, a combination of exercise and infrared therapy was used, and a reduction in IL-6 levels was observed, exerting an anti-inflammatory effect, and improving pain and quality of life in FM subjects [19]. However, after a 15-week strength exercise intervention, no anti-inflammatory effect was found on any FM symptoms [21]. From the methodological point of view of the meta-analysis, only a few studies could be analyzed. When the data distribution is symmetrical, the median can be used in meta-analyses. This criterion was adopted methodologically with the studies by Enberg et al. [17,20], so it would have been indicated to meta-analyze these studies using the probability of superiority (PS) as an index of effect size given the non-parametric analysis used (Mann-Whitney U test) [31,32]. Future lines of research could aim to carry out an analysis of the real impact of the size of the effect that is unknown by calculating the non-overlapping percentage of the population in order not to adopt risks. Likewise, in future research, it would be interesting to carry out a meta-regression that considered cytokines and the MFI (0–20) fatigue scale [33].

An important point of discussion that could be taken up in the future would be the probable association of virus infection with cytokines levels or immune responses in patients with FM. However, among the interests of the present study was to analyze the presence of anti-inflammatory and proinflammatory cytokines, but not to relate it to the probable viral association, since this hypothesis has been dismissed on some occasions. For example, what could be analyzed in the future is the possible relationship of FM symptoms with COVID-19 symptoms opening new etiopathogenetic horizons, but this objective will be pursued later.

The type of exercise, its duration, and intensity are also essential factors in the results obtained. For example, in a study of warm water aquatic exercise performed for eight months, two sessions per week, and 60 min per session, it was observed that after four months no anti-inflammatory effects were obtained. However, after eight months they were [15]. Although we know that all patients with chronic pain and FM have low-grade systemic inflammation, our meta-analysis was unable to demonstrate that blood biomarkers are specific or diagnostic for FM. The fact of having few studies and being very heterogeneous among themselves, does not provide sufficient scientific evidence to establish the cause-effect relationship of the treatments on the levels of cytokines and FM, for all this new research that analyzes the heterogeneity and seeks to carry out synthesis of results in a homogeneous way will be able to demonstrate the effect of the type of intervention carried out. In this case, it would be convenient to explore the moderate effect related to the participants and the characteristics of the intervention using regression techniques with confounding outcomes [11].

In another study of aquatic exercise in a heated pool, which lasted six weeks, three times per week and 50 min per session, together with thermotherapy, a decrease in IL-6 levels and improvements in pain and quality of life were observed in the FM group [19]. According to future research lines, we recommended examining continuous covariates that could be subjected to individual and pooled meta-regression in a random-effects meta-regression model taking into account the mean age of the women with FM, weight, height, study durations (weeks or months), sessions, days a week, number of exercises or treatment content, and even the method of blood extraction or cytokine measurement as predictor variables. Subgroups could also be used and analyzed for effects for categorical variables (women with fibromyalgia and healthy controls).

Therefore, although it was the same type of exercise, differences were observed in the duration of the intervention, the number of sessions per week, the duration of each session, and the addition of another physiotherapy tool.

## 5. Conclusions

In this systematic review, we found evidence to support elevated levels of proinflammatory and anti-inflammatory cytokines in patients with fibromyalgia compared to healthy subjects. These findings suggest the existence of chronic inflammation that may be responsible for an altered immune response and play a role in fibromyalgia symptoms, as well as developmental and nervous system sensitization to chronic pain. The most studied and analyzed cytokines were IL-1β, IL-6, IL-8, and TNF-α, while there was a lack of investigation of anti-inflammatory cytokines such as IL-4, IL-10, and IL-13. It was found through microanalyses that exercise affects the metabolic profile of muscle tissue but has no significant effect on the immune profile in the plasma of FM patients. Furthermore, results on the impact of exercise on cytokine levels have been variable, so it is essential to consider factors such as type of exercise, duration, intensity, and combination with other physiotherapeutic tools. These aspects are fundamental to formulating recommendations and intervention strategies for managing FM. However, our meta-analysis has reported a state-of-the-art, but the reported evidence is very low due to problems with the study design, the small number of participants, and the low certainty of the results after the effects of physiotherapy on cytokine levels.

## Figures and Tables

**Figure 1 biomedicines-11-02367-f001:**
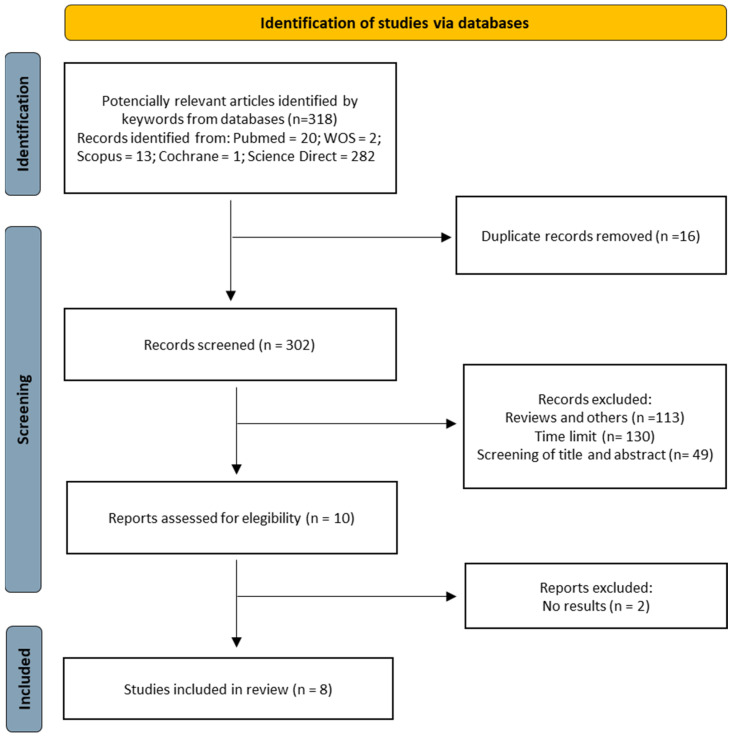
Flow diagram according to the PRISMA declaration.

**Figure 2 biomedicines-11-02367-f002:**
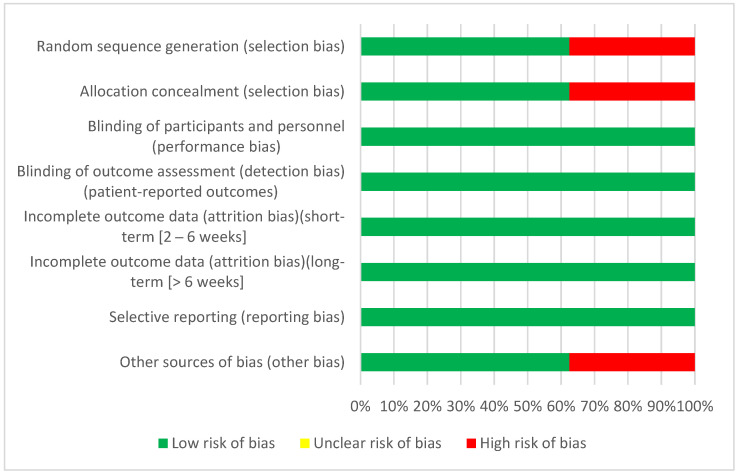
Risk of bias.

**Figure 3 biomedicines-11-02367-f003:**
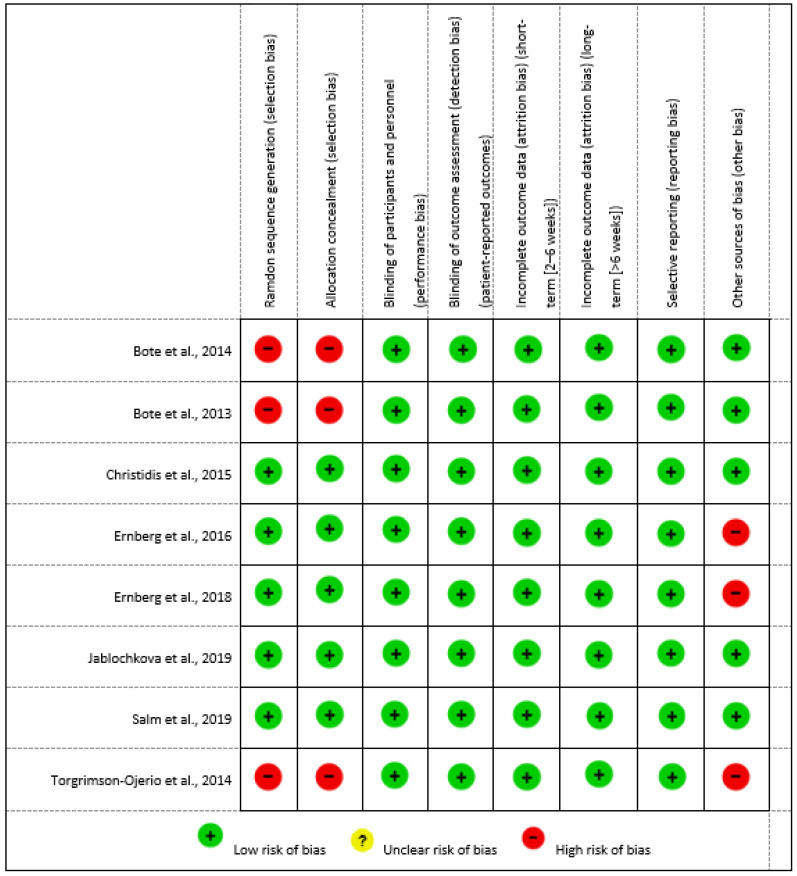
Risk of bias summary [15,16,17,18,19,20,21,22].

**Figure 4 biomedicines-11-02367-f004:**
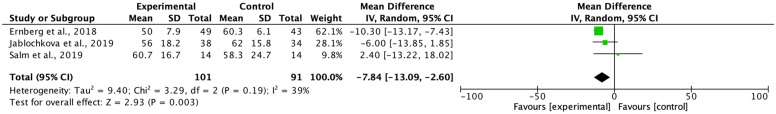
Effects of treatment versus the comparison group on the “Fibromyalgia Impact Questionnaire” (FIQ) [18,20,21].

**Figure 5 biomedicines-11-02367-f005:**
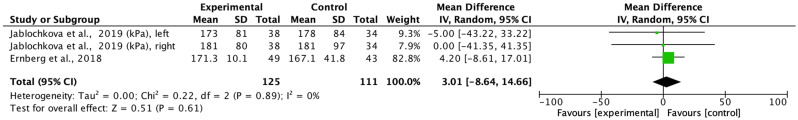
Effects of the treatment compared to the comparison group on the Algometry (PPT) in the left and right limb [20,21].

**Figure 6 biomedicines-11-02367-f006:**
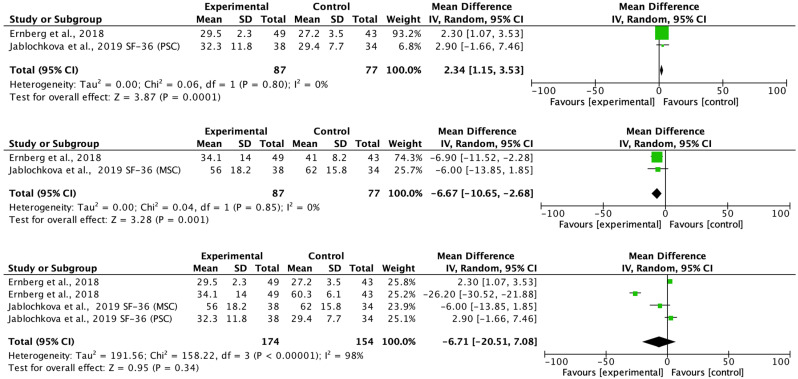
Effects of treatment versus comparison group on the QoL (SF-36-PSC, SF-36-MSC and combined effects) [20,21].

**Figure 7 biomedicines-11-02367-f007:**
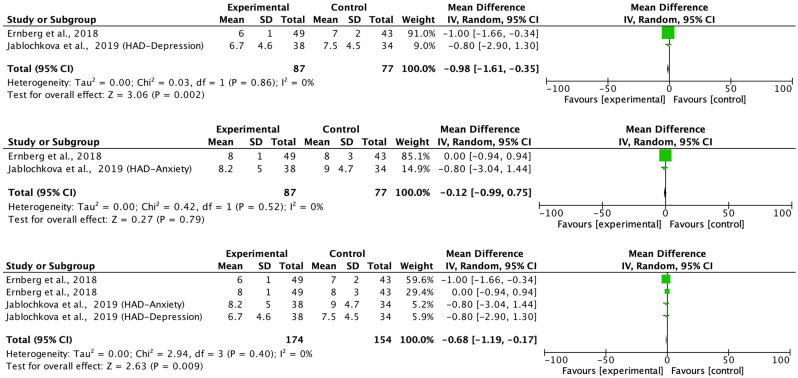
Effects of treatment versus comparison group on the “Anxiety and Depression Scale” Depression (HADS)-Depression, Anxiety, and combined effects [20,21].

**Figure 8 biomedicines-11-02367-f008:**
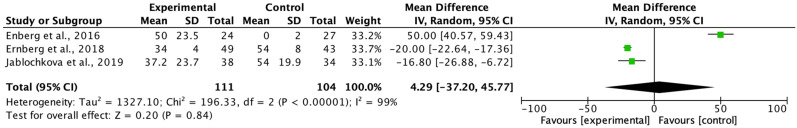
Effects of treatment versus comparison group on Pain (0–100) [20,21].

**Figure 9 biomedicines-11-02367-f009:**
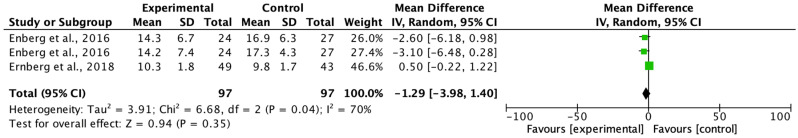
Effects of treatment versus comparison group on Static strength elbow flexion force (Kg)—Presented by right and left [17,20].

**Figure 10 biomedicines-11-02367-f010:**
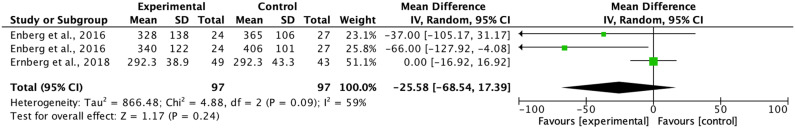
Effects of treatment versus comparison group on Static strength knee extension force (N)—Presented by right and left [17,20].

**Figure 11 biomedicines-11-02367-f011:**
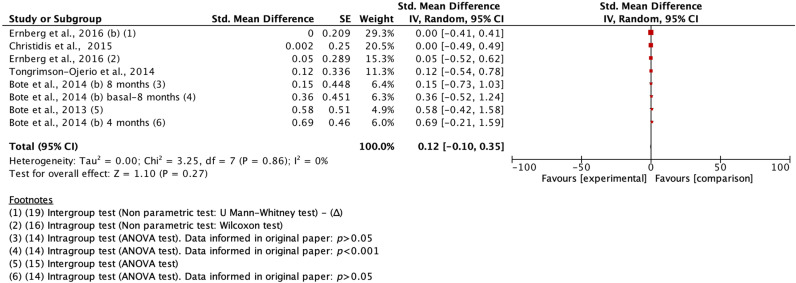
Effects of treatment versus comparison on IL-8 (pg/mL). b: second measurement of this outcome in the original study [15,16,17,19,22].

**Figure 12 biomedicines-11-02367-f012:**
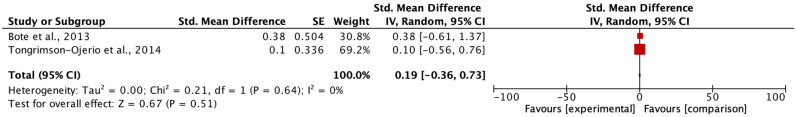
Effects of treatment versus comparison on Cortisol (pg/mL) [16,22].

**Figure 13 biomedicines-11-02367-f013:**
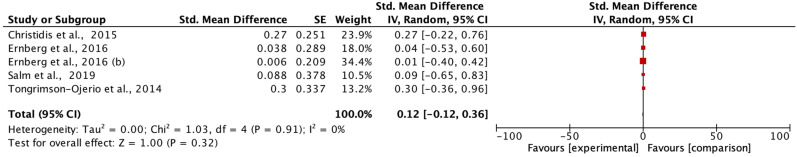
Effects of treatment versus comparison on TFN (pg/mL). b: second measurement of this outcome in the original study [17,18,19,21,22].

**Figure 14 biomedicines-11-02367-f014:**
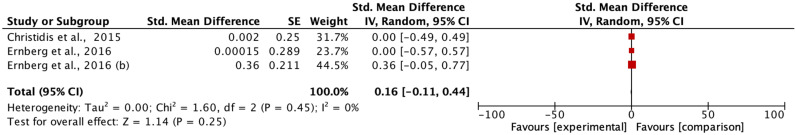
Effects of treatment versus comparison on IL-1β (pg/mL). b: second measurement of this outcome in the original study [17,19].

**Figure 15 biomedicines-11-02367-f015:**
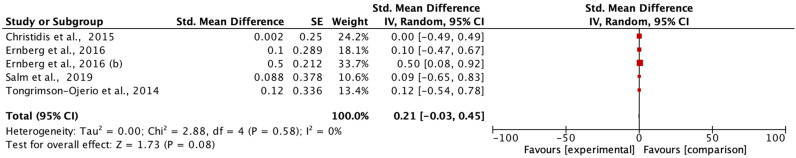
Effects of treatment versus comparison on IL-6 (pg/mL). b: second measurement of this outcome in the original study. b: second measurement of this outcome in the original study [17,18,19,22].

**Table 1 biomedicines-11-02367-t001:** Methodological assessment PEDro scale.

Reference	Type of Study	PEDro												Conflict of Interests
		1	2	3	4	5	6	7	8	9	10	11	TOTAL	
Bote et al., 2014 [15]	Clinical trial	+	−	−	+	−	−	−	+	+	+	+	5/10	N/A
Bote et al., 2013 [16]	Clinical trial	+	−	−	+	−	−	−	+	+	+	+	5/10	N/A
Christidis et al., 2015 [19]	Ramdomized clinical trial multicenter	+	+	+	+	+	+	+	+	+	+	+	10/10	N/A
Ernberg et al., 2016 [17]	Ramdomized clinical trial multicenter	+	+	+	+	+	+	+	−	−	+	+	8/10	NO
Ernberg et al., 2018 [20]	Ramdomized clinical trial multicenter	+	+	+	+	+	−	+	−	−	+	+	7/10	NO
Jablochkova et al., 2019 [21]	Ramdomized clinical trial	+	+	+	−	+	+	+	−	+	+	+	8/10	NO
Salm et al., 2019 [18]	Double-blind, ramdomized, placebo-controlled study	+	+	+	+	+	−	+	+	+	+	+	9/10	NO
Torgrimson-Ojerio et al., 2014 [22]	Clinical trial	−	−	−	+	−	−	−	+	+	+	+	5/10	N/A

1: elegibility criteria were specified; 2: subjects were ramdomly allocated to groups; 3: allocation was concealed; 4: the groups were similar at baseline regarding the most important prognostic indicators; 5: blinding of all subjects; 6: blinding of all therapist who administered the therapy; 7: blinding of all assessors who measured at least one key outcome; 8: >85% outcomes of the subjets initially allocated to groups; 9: data for at least one key outcome by “intention to treat”; 10: between-group statistical comparisons; 11: point measures and measures of variability; N/A: not available.

**Table 2 biomedicines-11-02367-t002:** GRADE system.

Author, Year and Reference	Type of Study	Evidence Quality
Bote et al., 2014 [15]	Clinical trial	Moderate
Bote et al., 2013 [16]	Clinical trial	Moderate
Christidis et al., 2015 [19]	Ramdomized clinical trial	High
Ernberg et al., 2016 [17]	Ramdomized clinical trial	High
Ernberg et al., 2018 [20]	Ramdomized clinical trial	High
Jablochkova et al., 2019 [21]	Ramdomized clinical trial	High
Salm et al., 2019 [18]	Ramdomized clinical trial	High
Torgrimson-Ojerio et al., 2014 [22]	Clinical trial	Moderate

## Data Availability

Data used for this manuscript are available on request.

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
