# Peer review of "The Effects of Non-Pharmacological Interventions in Fibromyalgia: A Systematic Review and Metanalysis of Predominants Outcomes"

_biomedicines, 2023, doi:10.3390/biomedicines11092367_

Round 1

Reviewer 1 Report

The authors of this meta-analysis wanted to analyze the cytokines levels in patients with FM compared to healthy individuals and also to examine the effect on cytokines levels of patients with FM exposed to different physiotherapy methods. The authors selected eight studies concluding that the systematic review found evidence of elevated cytokines levels in patients with FM and variable effects of physiotherapy interventions.

Major Issues

  • 1. The major flaws are the diversity of goals of the meta-analysis which lead to heterogeneous results. This heterogeneity was not recognized in the analysis and interpretation. Despite the limited number of articles required to draw a conclusion, the authors intended not only finding the presence of elevated cytokines, but also the type of cytokines associated with a FM in muscle and plasma and also the effects of different methods of physiotherapy on cytokines levels. There are challenges in detecting cytokines in different body fluids and multiple techniques, but without discussing the methodology for cytokine identification the results can be biased and the outcomes can be incorrect.
  • 2. Similar work about cytokines in fibromyalgia has already been published without the authors acknowledging this. (Kumbhare, Dinesha et al: Potential role of blood biomarkers in patients with fibromyalgia: a systematic review with meta-analysis. PAIN 163(7):p 1232-1253, July 2022).
  • 3. It is already known that all the patients with chronic pain including FM have a low grade systemic inflammation but the previous systematic literature review and meta-analysis couldn`t support the notion that these blood biomarkers are specific biomarkers to or diagnostic of FM.
  • 4. Targeting the effect of multimodal rehabilitation on the cytokines levels in patients with FM should have been the main goal of the article. The authors mentioned in page 9 in the results something about cytokines but not strong enough to prove their case. No Forrest plot regarding cytokines and the correlation with exercise was found in the study and I consider this a major presentational problem.
  • 5. Unclear figures, from figure 4 to 10 all of them indicated the effects of physiotherapy on fibromyalgia patients. These figures have no relation with the discussion about cytokines or the aim of the paper

I suggest a major revision of the paper addressing only the effects of physiotherapy on cytokine levels. Appendix 1, 2, 3 should be included in the paper with assessed biomarkers, method and type of body fluid for identification. The conclusion and the title should be changed accordingly. I think the authors should acknowledge the low to very low evidence due to study design issues, the small number of participants, and low certainty of results after the effects of physiotherapy on cytokine levels.

Author Response

Response to reviewers' comments: Manuscript biomedicines-2516922 entitled “Cytokine levels in Fibromyalgia patients and the effects of physiotherapy tools: A systematic review and metanalysis."

We want to express our gratitude to the Journal Editor and the Reviewers for the time spent on our manuscript and for their helpful and constructive comments. 

We have addressed the points raised by the Reviewers in the response letter and changes have been highlighted (in red) in the manuscript. We believe that the manuscript has been tuned in the light of the suggested additions.

Reviewers Comments to Author:

Reviewer 1:

Comment 1:

The authors of this meta-analysis wanted to analyze the cytokines levels in patients with FM compared to healthy individuals and also to examine the effect on cytokines levels of patients with FM exposed to different physiotherapy methods. The authors selected eight studies concluding that the systematic review found evidence of elevated cytokines levels in patients with FM and variable effects of physiotherapy interventions.

  1. The major flaws are the diversity of goals of the meta-analysis which lead to heterogeneous results. This heterogeneity was not recognized in the analysis and interpretation. Despite the limited number of articles required to draw a conclusion, the authors intended not only finding the presence of elevated cytokines, but also the type of cytokines associated with a FM in muscle and plasma and also the effects of different methods of physiotherapy on cytokines levels. There are challenges in detecting cytokines in different body fluids and multiple techniques, but without discussing the methodology for cytokine identification the results can be biased and the outcomes can be incorrect.

The authors really appreciate this comment. Thanks for this comment. You are totally right. Following your comment, we have added a discussion about heterogeneity and the importance of suppressing it in future research (introduction and discussion), and we have highlighted the limitation of studies that did not offer sufficient information to carry out the desired meta-analysis model, evidencing said limitation. The barriers to conducting studies of methodological quality have also been reflected. Thank you very much for your clear explanation and for guiding our study methodologically. We agree with this comment. The use of global health indicators is a thesis that we have supported for ten years to carry out homogeneous interventions and reduce the clinical, statistical, and methodological heterogeneity identified in previous review works. However, we agree that other variables on the target population should have been chosen, and we have indicated this in the discussion and in the limitations of our study.

We have added the analysis report from methodology, and in the data analysis section and discussion:

"However, the heterogeneity associated with the processes, techniques and instruments for the detection of cytokines in different body fluids represents an underlying factor in the literature that, however, could not be grouped in this meta-analysis, hence the importance of using the mean difference technique. standardized in the meta-analysis. The study was unable to perform any Forrest plot regarding cytokines and the correlation with exercise in the study as the authors did not show sufficient data to calculate the effect size from the original data from the individual article. Instead, a qualitative description of the effects on these variables has been made."

Comment 2:

Similar work about cytokines in fibromyalgia has already been published without the authors acknowledging this. (Kumbhare, Dinesha et al: Potential role of blood biomarkers in patients with fibromyalgia: a systematic review with meta-analysis. PAIN 163(7):p 1232-1253, July 2022).

We agree with this comment. Thanks for the clear explanation and the useful references. They will be thoughtfully answered in this response letter. We have added a discussion about heterogeneity and the importance of eliminating it in future research, and we have highlighted the limitation of studies that did not offer sufficient information to perform the desired meta-analysis model. We have added the reference indicating, the work of Kumbhare et al. (2021) as reference number 28 and we have started from their previous analysis, as can be read below in the title, abstract, introduction and discussion section our purpose according to our study object and the principal aim:

New tittle:

"Multimodal rehabilitation on cytokine levels on Fibromyalgia patients: A systematic review and metanalysis of predominant outcomes"

Introduction section:

"Recent reviews with metanalysis [28] revealed that individual biomarkers may play a relevant role in identifying pathologies coexisting with FM. However, new research could make possible to offer the predominant instruments and parameters depending on the type of intervention carried out. Once these parameters were classified, a grouping of predominant outcomes was carried out from the original data of the articles whenever these data were offered. The main aim of the meta-analysis was to examine the effect of multimodal rehabilitation on cytokine levels and other predominant variables in patients with FM."

Discussion section:

"Although we know that all patients with chronic pain and FM have low-grade systemic inflammation, our meta-analysis was unable to demonstrate that blood biomarkers are specific or diagnostic for FM. The fact of having few studies and very heterogeneous among themselves, does not provide sufficient scientific evidence to establish the cause-effect relationship of the treatments on the levels of cytokines and FM, for all this new research that analyzes the heterogeneity and seeks to carry out synthesis of results in a homogeneous way will be able to demonstrate the effect of the type of intervention carried out. In this case it would be convenient to explore the moderate effect related to the participants and the characteristics of the intervention using regression techniques with confounding outcomes [28]."

In our research a differentiation is made between proinflammatory and anti-inflammatory cytokines, in addition to having found in the present meta-analysis findings that suggest the existence of a chronic inflammation that could be the cause of an altered immune response, which could play a relevant role in the symptoms of fibromyalgia, in addition to a sensitization of the nervous system and the possible development of pain.

In addition, the present review study has contemplated the investigation and differentiation of cytokines present in the muscle tissue and plasma of patients with FM, an aspect not contemplated in other studies.

Comment 3:

It is already known that all the patients with chronic pain including FM have a low grade systemic inflammation but the previous systematic literature review and meta-analysis couldn`t support the notion that these blood biomarkers are specific biomarkers to or diagnostic of FM.

The authors really appreciate this comment. We agree with this comment. Indeed, the protocol carried out during the intervention is essential for the control and reproducibility of the study.

Unfortunately, we don´t have adequate information to carry out the meta-analysis with the rest of the variables. This is due to the fact that in the articles we do not have the adequate information to calculate the size of the effect (they do not offer means and standard deviations and, in several cases, they only report the results through figures in which the exact value of descriptive statistics).

In other studies, the median and interquartile range are shown as descriptive values, so they cannot be included in the model either (this is the case of the two studies by Enberg et al.). They have not measured all the variables in this way. Why did I use this procedure with some and not with others? It is a correct procedure: this is probably due to the fact that the variables examined and presented with the means and the standard deviation were adjusted to the Normal Distribution; however, when this is not the case, it is advisable to use the median and the interquartile range because it is not possible to characterize the distribution of the population from which the sample is made up, hence the authors have adopted this criterion. It's a shame, especially to assess the effect on cytokines, quality of life, anxiety and depression.

Despite the fact that not all the studies median the same variables (see Annex 4 of the paper, p. 32) we have managed to examine three studies in three different variables: FIQ, PPT and HADS, so I suggest directing this first meta-analysis study to the perception of symptoms in fibromyalgia, pain and anxiety. In any case, I think it is very good: despite the fact that there are many instruments, some measure the same variable but in a different way (VAS, EVA, Algometry, etc.), which is a point of discussion regarding the heterogeneity of the measurement and the need to standardize the analysis using the standardized mean difference (SMD) procedure, since the instruments measured different scales, which could suggest its use (Jonathan J Deeks, Julian PT Higgins, Douglas G Altman; Chapter 6, Section 6.5.1.2).

The standard error is the standard deviation (as in the study by Torgimson-Ojeiro et al.)

In the studies by Bote et al. (2013, 2013b) the meta-analysis model cannot be implemented on any variable.

In conclusion:

Given the complexity and breadth of this work, one way of taking advantage of this meta-analysis is that we divide it into two (separately, they would be two contiguous publications). In a first publication I suggest publishing the meta-analysis that we present now; and, for later, propose a second meta-analysis from the calculation of the effect size separately in another software that allows us to manually enter the d or Eta squared, and that contemplates the following variables (1) and techniques (2):

  1. Clinical analysis of cytokines
  2. Meta-regression

In this case I suggest exploring the moderate effect related to the participants and the characteristics of the intervention. To do this, in a second meta-analysis we can perform a meta-regression. That is, we can subject continuous covariates to individual and joint meta-regression in a random-effects meta-regression model (using SPSS). We can take into account the following predictor variables: average age of women with FM, weight, height, study durations (weeks or months), sessions, days a week, number of exercises or treatment content, and even the method of blood extraction or measurement of cytokines, etc. We can also use subgroups and analyze them for effects for the effects of categorical variables (women with fibromyalgia and healthy controls).

Reference:

Higgins J.P., Thompson S.G., Deeks J.J., Altman D.G. Measuring inconsistency in meta-analyses. BMJ. 2003;327:557–560. doi: 10.1136/bmj.327.7414.557. 

Comments 4 and 5:

  1. Targeting the effect of multimodal rehabilitation on the cytokines levels in patients with FM should have been the main goal of the article. The authors mentioned in page 9 in the results something about cytokines but not strong enough to prove their case. No Forrest plot regarding cytokines and the correlation with exercise was found in the study and I consider this a major presentational problem.

  1. Unclear figures, from figure 4 to 10 all of them indicated the effects of physiotherapy on fibromyalgia patients. These figures have no relation with the discussion about cytokines or the aim of the paper.

Thank you very much for this rigorous and pertinent comments. Unfortunately, we cannot carry out the statistical exploitation of cytokines and the correlation with exercise because the authors did not report measures of scattered tendency such as SD, and often the data were endorsed with figures or ranges that make the analysis more complex (there are solutions in the literature in order to homogenize such analyses, I insert some references, however, this would require a much deeper approach that extends the time devoted to this review):

  • Higgins JPT, Thomas J, Chandler J, et al. Cochrane handbook for systematic reviews of interventions version 6.3. February, 2022. www.training.cochrane.org/handbook (accessed March 12, 2023).
  • Wan X, Wang W, Liu J, Tong T. Estimating the sample mean and standard deviation from the sample size, median, range and/or interquartile range. BMC Med Res Method 2014; 14:135.
  • Smart NA, Waldron M, Ismail H, et al. Validation of a new tool for the assessment of study quality and reporting in exercise training studies: TESTEX. Int J Evid-Based Healthc 2015; 13:9–18.
  • Seide SE, Röver C, Friede T. Likelihood-based random-effects meta-analysis with few studies: empirical and simulation studies. BMC Med Res Method 2019; 19:16.
  • Dias S, Welton NJ, Caldwell DM, Ades AE. Checking consistency in mixed treatment comparison meta-analysis. Stat Med 2010; 29:932–44.

To counteract this fact, I have tried to incorporate some of the reviewer's suggestions in the title and the aim, qualifying their application in the discussion, acknowledging of course the limitations of this study and carrying out, in line with the aforementioned work by Kumbhare et al, future lines of research.

We have specified this information in our discussion and conclusions to be humble and offer the scientific community a rigorous state of the art that adjusts to our poor findings:

Discusion section:

"According to future research lines, we recommended to examine continuous covariates could be subjected to individual and pooled meta-regression in a random-effects meta-regression model taking into account the mean age of the women with FM, weight, height, study durations ( weeks or months), sessions, days a week, number of exercises or treatment content, and even the method of blood extraction or cytokine measurement as predictor variables. Subgroups could also be used and analyzed for effects for categorical variables (women with fibromyalgia and healthy controls).

Conclussion section:

"However, our meta-analysis has reported a state of the art, but the reported evidence is very low due to problems with study design, the small number of participants, and the low certainty of the results after the effects of physiotherapy on cytokine levels."

Unfortunately, we have hardly been able to work with two studies, since if there was no problem, there was another. That is, if the study measured the variable, it did not offer the information we needed (many of them report using figures and inferences, not offering the descriptive statistics necessary to calculate the size of the effect); when we did not find precisely the opposite, a study that measured other variables that did not measure the rest. Although there are studies that measured (up to five times) the same variable, the truth is that with the data provided we cannot calculate the model. However, in these observations I propose possible ways of working and discussing. Likewise, Appendices 1, 2, 3 have been included in the document with the biomarkers evaluated, the method and the type of body fluid for identification.

Reviewer 2 Report

This systemic review and meta-analysis aim to analyze the available evidence of cytokine levels in patients with fibromyalgia (FM) compared to healthy subjects. Although this is an important issue and a comprehensive review, some comments need to be addressed as follows:   

Major comments

1. It would be better to show the table with list of studies indicating the proinflammatory cytokines and anti-inflammatory cytokines in patients with FM for readers. Besides, it would be helpful to illustrate in a figure, which reveals the involvement of cytokines in the pathogenesis of FM.

2.   Could you provide a little insight into any animal models? I know that data regarding cytokines and immune responses in animal models are scarce, but this could complete your manuscript.

3.    The role of central cytokines and innate immunity seems to be at the center of pathophysiology of FM as the name of title. However, it seems that the focus becomes diffuse and shows the effects of treatment on the FIQ, the algometry, QoL,…. Thus, the work would follow the title and the aims of this systemic review.

4.    It would be interesting that the authors highlight the probable association of virus infection with cytokines levels or immune responses in patients with FM. This would enrich this review.

5. Given an important role of gender in the pathogenesis of FM, the authors would have addressed the differences in cytokine levels between male patients and female patients with FM.

Minor English editing is needed.

Author Response

Response to reviewers' comments: Manuscript biomedicines-2516922 entitled “Cytokine levels in Fibromyalgia patients and the effects of physiotherapy tools: A systematic review and metanalysis."

We want to express our gratitude to the Journal Editor and the Reviewers for the time spent on our manuscript and for their helpful and constructive comments. 

We have addressed the points raised by the Reviewers in the response letter and changes have been highlighted (in red) in the manuscript. We believe that the manuscript has been tuned in the light of the suggested additions.

Reviewers Comments to Author:

Reviewer 2:

This systemic review and meta-analysis aim to analyze the available evidence of cytokine levels in patients with fibromyalgia (FM) compared to healthy subjects. Although this is an important issue and a comprehensive review, some comments need to be addressed as follows:

Major comments

Comment 1:

  1. It would be better to show the table with list of studies indicating the proinflammatory cytokines and anti-inflammatory cytokines in patients with FM for readers.

Besides, it would be helpful to illustrate in a figure, which reveals the involvement of cytokines in the pathogenesis of FM.

The authors really appreciate this comment. Thanks for this comment. You are totally right. We have added an appendix (appendix 5), where we have included the involvement of cytokines in the pathogenesis of FM.

Comment 2:

  1. Could you provide a little insight into any animal models? I know that data regarding cytokines and immune responses in animal models are scarce, but this could complete your manuscript.

Thank you very much for this rigorous and pertinent comments. Unfortunately, the present research does not contemplate the objective of analyzing animal models. However, in the search performed prior to this research, the keywords "cytokines" AND fibromyalgia AND animal models were included, and no results were found.

Comment 3:

  1. The role of central cytokines and innate immunity seems to be at the center of pathophysiology of FM as the name of title. However, it seems that the focus becomes diffuse and shows the effects of treatment on the FIQ, the algometry, QoL,…. Thus, the work would follow the title and the aims of this systemic review.

Thank you very much for your rigorous and pertinent comment. We fully agree and, so that it can better illustrate the purpose of this research considering the meta-analytic model that we have finally been able to apply, the authors have proposed this title.

New tittle:

"Multimodal rehabilitation on cytokine levels on Fibromyalgia patients: A systematic review and metanalysis of predominant ouctomes"

Comment 4:

  1. It would be interesting that the authors highlight the probable association of virus infection with cytokines levels or immune responses in patients with FM. This would enrich this review.

The authors really appreciate this comment. Among the interests of the present study was to analyze the presence of anti-inflammatory and proinflammatory cytokines, but not to relate it to the probable viral association, since this hypothesis has been dismissed on some occasions. For example, what could be analyzed in the future is the possible relationship of FM symptoms with COVID-19 symptoms opening new etiopathogenetic horizons, but this objective will be pursued later. We have added this content in the discussion section.

Comment 5:

  1. Given an important role of gender in the pathogenesis of FM, the authors would have addressed the differences in cytokine levels between male patients and female patients

with FM.

The authors really appreciate this comment. Unfortunately, the objective of the present study is to analyze women with FM, since at present the rate of diagnosis is still higher in women than in men, which undoubtedly represents a limitation to be able to draw conclusions about this disease in the male gender.

Reviewer 3 Report

The article is well written. I would like to recommend its publication. I have several suggestions for this article:

First, the introduction can be more concentrated. The general information of fibromyagia can be shortened.

Second, to brief fibromyagia, the annotation that includes “FM” should be parentheses.

Third, in line 95, the database for registration and the registration number should be provided.

Fourth, the exact PICO question should be spelt out.

Fifth, the resolution of the forest plots should be improved.

Author Response

Response to reviewers' comments: Manuscript biomedicines-2516922 entitled “Cytokine levels in Fibromyalgia patients and the effects of physiotherapy tools: A systematic review and metanalysis."

We want to express our gratitude to the Journal Editor and the Reviewers for the time spent on our manuscript and for their helpful and constructive comments. 

We have addressed the points raised by the Reviewers in the response letter and changes have been highlighted (in red) in the manuscript. We believe that the manuscript has been tuned in the light of the suggested additions.

Reviewers Comments to Author:

Reviewer 3:

The article is well written. I would like to recommend its publication. I have several suggestions for this article:

First, the introduction can be more concentrated. The general information of fibromyagia can be shortened.

Thank you very much for this comment. We have changed the introduction as the reviewer requested.

Second, to brief fibromyagia, the annotation that includes “FM” should be parentheses.

Thank you very much for this comment. We have changed in the introduction the term [FM] to (FM) as requested.

Third, in line 95, the database for registration and the registration number should be provided.

The authors really appreciate this comment. We agree with this comment. We have included the protocol registered number as requested.

Fourth, the exact PICO question should be spelt out.

The authors really appreciate this comment. We agree with this comment. We have included the following paragraph:

“P - patients with FM according to ACR 1990; I - physiotherapy intervention tool; C - comparison of healthy group vs FM group (same intervention). FM group vs FM group (physiotherapy and relaxation intervention); O - results...levels of proinflammatory and anti-inflammatory cytokines. In control group and experimental group”.

Fifth, the resolution of the forest plots should be improved.

The authors really appreciate this comment. We agree with this comment. The forest plots have been improved as the reviewer requested.

Round 2

Reviewer 1 Report

Thank you for answering my questions. I still have some issues with the aim of the paper, the results and the discussions. As I mentioned before [ in point 5] the figures 4,5,6,7,8,9 10 all of them indicated the effects of treatment on fibromyalgia patients. These figures have no relation with the discussion about cytokines or the aim of the paper. I could not see what figures were deleted and which figures were inserted in the document.

Minor: 

page 14 from 510-518 I do not understand the speculation about Covid-19 and fibromyalgia,

Author Response

Response to reviewers' comments: Manuscript biomedicines-2516922 entitled “Cytokine levels in Fibromyalgia patients and the effects of physiotherapy tools: A systematic review and metanalysis."

We want to express our gratitude to the Journal Editor and the Reviewers for the time spent on our manuscript and for their helpful and constructive comments. 

We have addressed the points raised by the Reviewers in the response letter and changes have been highlighted (in red) in the manuscript. We believe that the manuscript has been tuned in the light of the suggested additions.

Reviewers Comments to Author (Round 2):

Reviewer 1:

Comment 1:

Thank you for answering my questions. I still have some issues with the aim of the paper, the results and the discussions. As I mentioned before [ in point 5] the figures 4,5,6,7,8,9 10 all of them indicated the effects of treatment on fibromyalgia patients. These figures have no relation with the discussion about cytokines or the aim of the paper. I could not see what figures were deleted and which figures were inserted in the document.

The authors really appreciate your comment. We fully agree with your comment. The authors consider the same in relation to the results about cytokines and their relationship with the objective and discussion with the article. Despite the limitations noted in our previous review, our research group put a lot of effort into performing a quantitative analysis of the results in a new meta-analysis model on the variables IL-8, Cortisol, TNF, IL-1β, and IL-6. Please see the changes written in red color. As the authors of the original included trials did not report the data we needed to perform the meta-analysis or we found high statistical heterogeneity, these have had to be examined according to a rigorous calculation process. The methodology used can be read in lines 241-259:

"For the calculation of the effect size in cytokines, the predominant cytokine parameters at plasma levels (pg/mL) were selected and pooled from the original data of the different trials when a minimum N was obtained to two studies. Several trials reported median and interquartile ranges in their study.To combine the results of these studies, the mean and standard deviation of the sample were estimated using the method of Wan et al.[29].This method, less effective that the method proposed by Hozo et al. al [30], was more suitable for such an estimate.In cases where studies reported only p value greater than 0.05 or mean difference (Δ) in inter- and intragroup pre- and post-test value change, estimated values were taken from the original study data.Once all the necessary scores were obtained, the effect size of each parameter was calculated according to the method proposed by Lipsey and Wilson [31].When the hypothesis testing of the original trial employed nonparametric techniques or more than two means were compared, Eta squared or Eta squared partial (η2) was calculated directly or from Cohen's d using the method of Lenhard et al. [32]. In some cases, the z-contrast statistic was calculated from the p-value offered in the trial. The cytokine meta-analysis model was random-effects based on the standardized mean difference and standard error of each study, also previously calculated [31]. The size of the effect on cytokines was considered small around 0.01, about 0.06 was considered a medium effect and greater than 0.14 a large effect, this being negligible when 0 was found in the CI [33]."

We agree with all your observations. The interest of the work was to know the effect of the different interventions in patients with FM on cytokines, for this reason I conformed the meta-analysis model on most of these parameters in all the studies. Finally we have added five more figures in the results (Figures 11-15). As we said, in the results section we present the results of our analysis on these new cytokine variables and in the methodology section I have incorporated the procedure followed.

Similarly, I have expanded and supplemented the discussion based on this reviewer's observations and added the cited references.

Discussion

Lines 477-486:

"…The intervention by Ernberg et al. (19) significantly decreased the scores of the FIQ in favor of the relaxation group, but the two fibromyalgia groups examined did not find functional and clinical differences in the change in values from pre to post-test. Other studies showed differences in variables in favor of the control group in pain ratios applying programs of 15 weeks of strength exercise 2 sessions/week 1 session/60min 10 min warm-up 50 min and lower limb strength, however, this intervention did not normalize a chronic inflammation profile nor did it have any effect on the anti-inflammatory effect in patients with FM symptoms on the clinical and functional variables examined (16, 19, 20)."

Lines 499-509:

"Although the heterogeneity associated with the processes, techniques, and instruments for the detection of cytokines in different body fluids represents an underlying factor in the literature, we highlight the importance of using the mean difference technique. standardized in the meta-analysis. The study had difficulties in making Forrest plots regarding cytokines and the correlation with exercise in the study, as the authors did not show sufficient data to calculate the effect size from the original individual trial data. Our work showed a high methodological and statistical heterogeneity of the studies. The lack of data made it impossible to analyze all the variables, which is a limitation when it comes to quantifying the effects. This could affect the results of the meta-analysis, as some effects found were negligible in the meta-analytic model. However, a qualitative description of the effects on these variables has been made."

References added in the methodology:

  1. Wan, X., Wang, W., Liu, J. et al. Estimating the sample mean and standard deviation from the sample size, median, range and/or interquartile range. BMC Med Res Methodol 14, 135 (2014). https://doi.org/10.1186/1471-2288-14-135

  1. Hozo, S.P., Djulbegovic, B. & Hozo, I. Estimating the mean and variance from the median, range, and the size of a sample. BMC Med Res Methodol 5, 13 (2005). https://doi.org/10.1186/1471-2288-5-13

  1. Lipsey, Mark W., and David B. Wilson. Practical meta-analysis. SAGE publications, Inc, 2001.

  1. Lenhard, W. & Lenhard, A. (2016). Computation of effect sizes. Retrieved from: https://www.psychometrica.de/effect_size.html. Psychometrica. I: 10.13140/RG.2.2.17823.92329

  1. Cohen, J. Statistical Power Analysis for the Behavioral Sciences (2nd ed.). (1988). Lawrence Erlbaum Associates, Publishers

Comment 2:

Minor: page 14 from 510-518 I do not understand the speculation about Covid-19 and fibromyalgia.

Thank you very much for your comment, the authors really appreciate coming back to this point. This has been the subject of scientific discussion in our group. The text is since Reviewer 2 in the first review process made the following comment: "It would be interesting that the authors highlight the probable association of virus infection with cytokines levels or immune responses in patients with FM. This would enrich this review." Our team reasoned about the possible link with infectious processes and Reviewer 2 agreed with the text included in the second round of review, hence the preservation of this idea. While it is still necessary to expand the scientific literature around the new Covid-19 disease, our group considered it interesting to open a possible line of research or discussion in the association of variables that could predict scenarios or relate clinical mechanisms between an infection and clinical and functional variables.

Reviewer 2 Report

The authors have addressed all my queries. I do not have any further comment. 

Author Response

Response to reviewers' comments: Manuscript biomedicines-2516922 entitled “Cytokine levels in Fibromyalgia patients and the effects of physiotherapy tools: A systematic review and metanalysis."

We want to express our gratitude to the Journal Editor and the Reviewers for the time spent on our manuscript and for their helpful and constructive comments. 

We have addressed the points raised by the Reviewers in the response letter and changes have been highlighted (in red) in the manuscript. We believe that the manuscript has been tuned in the light of the suggested additions.

Reviewers Comments to Author (Round 2):

Reviewer 2:

Comments:

The authors have addressed all my queries. I do not have any further comment. 

The authors really appreciate your time and dedication spent on our manuscript and are honored by the important observations you have made, which have enriched our manuscript and contribution.

Round 3

Reviewer 1 Report

Thank you for the improvement of the manuscript.

1. The title should be changed in
The effects of non-pharmacological interventions in fibromyalgia. A systematic review and metanalysis of predominants outcomes

2. The reference in the text and in the list should be changed accordingly

Author Response

Response to reviewers' comments: Manuscript biomedicines-2516922 entitled "The effects of non-pharmacological interventions in fibromyalgia. A systematic review and meta-analysis of the prevailing results"

We would like to express our gratitude to the editor of the journal and to the reviewers for the time dedicated to our manuscript and for their useful and constructive comments.

Reviewers' comments to the author (Round 3):

Reviewer 1 (Answer):

The authors really appreciate your kind dedication in reviewing our manuscript again and providing your valuable comments. The authors agree with you on your title proposal, which we consider very adequate and pertinent with respect to the objective and content of our research. In the same way, we have attended to their considerations in the manuscript. Please see attached the manuscript with your learned considerations. Thanks again for your time and great suggestions.